# Quercetin Protects against Lipopolysaccharide-Induced Intestinal Oxidative Stress in Broiler Chickens through Activation of Nrf2 Pathway

**DOI:** 10.3390/molecules25051053

**Published:** 2020-02-26

**Authors:** Lei Sun, Gaoqing Xu, Yangyunyi Dong, Meng Li, Lianyu Yang, Wenfa Lu

**Affiliations:** College of Animal Science and Technology, Jilin Agricultural University, Changchun 130118, China; jzyxysl@163.com (L.S.); xugaoqing1995@163.com (G.X.); dongyangyunyi@163.com (Y.D.); limengtudou@163.com (M.L.)

**Keywords:** quercetin, oxidative stress, gut, broiler chickens, Nrf2, MAPK

## Abstract

We investigated the potential ability of quercetin to protect against lipopolysaccharide (LPS)-induced intestinal oxidative stress in broiler chickens and the potential role of the Nrf2 (nuclear factor erythroid 2-related factor 2) signaling pathway. One-day-old broiler chickens (n = 240) were randomized into four groups: saline-challenged broiler chickens fed a basal diet (Con), LPS-challenged broiler chickens on a basal diet (LPS), and LPS-treated broiler chickens on a basal diet containing either 200 or 500 mg/kg of quercetin (Que200+LPS or Que500+LPS). Quercetin (200 mg/kg) significantly alleviated LPS-induced decreased duodenal, jejunal, and illeal villus height and increased the crypt depth in these regions. Quercetin significantly inhibited LPS-induced jejunal oxidative stress, including downregulated reactive oxygen species (ROS), malondialdehyde (MDA), and 8-hydroxy-2′-deoxyguanosine (8-OHdG) levels, and it upregulated superoxide dismutase (SOD) and glutathione peroxidase (GSH-Px) levels. Quercetin relieved LPS-induced jejunal mitochondria damage and upregulated mitochondrial DNA copy number-related gene expression, including cytochrome c oxidase subunit 1 (COX1), ATP synthase F0 subunit 6 (ATP6), and NADH dehydrogenase subunit 1 (ND1). Quercetin attenuated the LPS-induced inhibition of Nrf2 activation, translocation, and downstream gene expression, including heme oxygenase-1 (HO-1), NAD (P) H dehydrogenase quinone 1 (NQO1), and manganese superoxide dismutase (SOD2). Additionally, quercetin attenuated the LPS-inhibition of c-Jun N-terminal kinase (JNK), Extracellular Regulated protein Kinases (ERK), and p38MAPK (p38) phosphorylation in the MAPK pathway. Thus, quercetin attenuated LPS-induced oxidative stress in the intestines of broiler chickens via the MAPK/Nrf2 signaling pathway.

## 1. Introduction

Bacterial infection is a major cause of intestinal oxidative stress [1]. In broiler chickens, bacterial infections seriously affect intestinal health and growth performance [2]. Lipopolysaccharide (LPS), a Gram-negative bacteria cell wall component, is an effective biostimulant for the immune system. Additionally, LPS could induce reactive oxygen species (ROS) production and damage to lipids, producing a biomarker of lipid degradation (malondialdehyde, MDA) [3,4,5]. Further, LPS could regulate intestinal oxidative status, leading to intestinal injury in broiler chickens [6]. To prevent bacterial infection, antibiotics have been widely used in the broiler industry, which has caused more prominent problems, such as bacterial resistance [7]. Therefore, it is very important to replace antibiotics with an effective feed additive that can protect broiler chickens from intestinal oxidative stress.

Quercetin (3,3′,4′,5,7-pentahydroxyflavone) is a flavonoid that is widely found in several foods, including grains, fruits, vegetables, and tea [8]. Quercetin has anti-inflammatory, anti-oxidant, and anti-apoptosis properties and thus, it recommended as a nutritional supplement for animals [9,10,11,12]. Moreover, quercetin can prevent oxidative stress by scavenging free radicals, removing oxidation products, and stimulating antioxidant enzymes in many animal models [13,14]. However, its effect on intestinal oxidative stress in broiler chickens is still unclear.

The redox-sensitive nuclear factor erythroid 2-related factor 2 (Nrf2) transcription factor is the main defense mechanism against various harmful stresses; it improves the body’s antioxidant status and maintains cellular redox homeostasis [15,16]. Nrf2 can regulate downstream antioxidants to exert stronger protective effects, including heme oxygenase-1 (HO-1), NAD(P)H dehydrogenase quinone 1 (NQO1), and superoxide dismutase (SOD) [17,18]. However, the involvement of Nrf2 signaling in the effects of quercetin on intestinal oxidative stress in broiler chickens is yet to be verified.

Therefore, we hypothesized that quercetin could alleviate LPS-induced intestinal oxidative stress in broiler chickens via the Nrf2 signaling pathway.

## 2. Materials and Methods

### 2.1. Animals Treatment

Ethical approval for the present study was obtained from the Ethical Committee of the Jilin Agricultural University, China (20171832). In total, 240 one-day-old male broiler chickens (Arbor Acres) of similar body weight were selected from the broiler farm in Dehui City, Jilin Province, China. During the experiment, all chickens were placed in a climate-controlled room. From day 1 to day 7, the temperature of the room was maintained at 35 °C; after that, the temperature was reduced by 2 °C every week until it reached 24 °C.

The chickens were randomized into 4 treatment groups with 6 replicate pens of 10 broiler chickens per pen: (1) basal diet + saline (control group; ‘Con’ hereafter); (2) basal diet + LPS (LPS); (3) basal diet + 200 mg/kg Que + LPS (Que200+LPS); and (4) basal diet + 500 mg/kg Que + LPS (Que500+LPS). The basal diet, shown in Table 1, was mixed and provided as mash. Quercetin (98%, Changyue biological techology, Xi’an, China) was mixed in the basal diet. On day 16, 18, and 20 of the 21-day feeding trial, broiler chickens in the LPS, Que200+LPS, and Que500+LPS groups received an intraperitoneal LPS challenge (E. coli serotype 055: B55; Sigma, Saint Louis, MO, USA). The LPS injection volume was 0.5 mg/kg bodyweight (diluted with sterilized saline to 0.5 mg/mL) at 500 μg/mL/kg dissolved in sterile saline. Con group animals received an equivalent amount of sterile saline [19]. At 21 d of age, 6 broiler chickens randomly selected from each group were slaughtered and samples were collected.

### 2.2. Sample Collection

On day 21, 6 chickens were randomly selected from each group and euthanized. They were necropsied with the small intestine being exposed and separated from the mesentery. Intestinal samples (about 4 cm segments) were collected from three small intestine sites: the duodenum (from the gizzard outlet to the end of the pancreatic loop), the jejunum (from the pancreatic loop to Meckel’s diverticulum), and the ileum (from Meckel’s diverticulum to the cecal junction). Each intestinal sample was divided into two parts: one part was washed using ice cold PBS (pH 7.4) followed by fixation using 10% formalin solution, and the other part was placed in liquid nitrogen after washing. To avoid cross contamination, we first collected Con group samples, followed by sampling from the challenged treatment groups.

### 2.3. Intestinal Morphology

After sample dehydration and paraffin embedding, 4-μm sections of the duodenum, jejunum, and ileum tissue from each treatment group were prepared and hematoxylin/eosin stained. Then, microscopic imaging was then used to determine the villus height and crypt depth in each sample via a computer-assisted morphometric system (Nikon, Tokyo, Japan). Villi height was measured as the length from the villi tip to base, while the crypt depth was the length from the villi base to crypt base.

### 2.4. Measurement of Antioxidant Parameters

The jejunal tissue (0.1 g of each sample) was homogenized in PBS and then centrifuged at 2500 rpm for 10 min. The supernatants were stored at −20 °C for subsequent analysis. According to the manufacturers’ instructions (Shanghai Meilian Biology Technology, Shanghai, China), 10 μL of supernatant, 40 μL of sample diluent, and 100 μL of HRP-conjugate reagent were added to each antibody well and incubated at 37 °C for 1 h. After washing and chromogen, the ROS level was detected at 450 nm using a microplate reader. According to the instructions of ELISA kits (Shanghai Langdun Biotech, Shanghai, China), 50 μL of supernatant and 50 μL of biotin-labeled recognition antigen were added to each antibody well in triplicate and incubated at 37 °C for 30 min. After washing with PBST, avidin-HRP was added and incubated at 37 °C for 30 min. The chromogen solution and the stop solution were added to each well after washing again, and the levels of glutathione peroxidase (GSH-Px) and 8-hydroxy-2′-deoxyguanosine (8-OHdG) were detected at 450 nm using a microplate reader. Additionally, we used MDA, CAT, and SOD kits to detect and calculate the MDA, CAT, and SOD content by chemical colorimetry (Nanjing Jiancheng Bioengineering Institute, Nanjing, China).

### 2.5. Real-Time Quantitative PCR

From this section, samples in the Que500+LPS group were not used to determine the subsequent index. The LPS+Que group refers to the Que200+LPS group. TRIzol was used to extract the RNA of jejunal tissue, after which a PrimeScript RT reagent Kit with gDNA Eraser (Takara, Tokyo, Japan) was used to prepare DNA from 1 ug of the total sample RNA. RT-qPCR was conducted with a SYBR Premix Ex Taq II (Takara) using a Real-Time PCR Detection Platform (Agilent StrataGene Mx3005P, Santa Clara, CA, USA). Primer information for the relative genes and β-actin are shown in Table 2. The 2-∆∆Ct method was used to quantify gene expression.

### 2.6. Determination of mtDNA Copy Number and Transmission Electron Microscopy (TEM) Analysis

DNA of the jejunum from different groups was extracted according to the kit instructions (TIANGEN, Beijing, China). To quantify mtDNA copy number by real-time quantitative PCR, three fragments were targeted: ATP6 (ATP synthase F0 subunit 6), COX1 (cytochrome c oxidase subunit 1), and ND1 (NADH dehydrogenase subunit 1). A stable nuclear single copy gene, AGRT, served as the reference gene. All the primer sequences shown in Table 2 were designed and synthesized by Sangon Biotech (Shanghai, China). TEM analysis was performed to observe the ultrastructure of mitochondria. The Jejunum tissues were prepared by routine methods for TEM analysis. In brief, the samples were immersed in phosphate buffer containing 2.5% glutaraldehyde at 4 °C overnight and then subjected to dehydration and embedding. Sections were prepared in ultramicrotome (YESLAB, shanghai, China), and then, they were double-stained with uranyl acetate and lead citrate. Finally, the ultrastructure of mitochondria was observed with electron microscopy (JEOL, Tokyo, Japan).

### 2.7. Western Blot Analysis

RIPA Lysis Buffer (Beyotime Institute of Biotechnology, Shanghai, China) was used to extract the total protein of jejunal tissue, with a BCA kit (Beyotime Institute of Biotechnology) then being used to quantify sample protein amounts. Then, protein was separated via SDS-PAGE prior to transfer onto nitrocellulose membranes (Merck Millipore, Darmstadt, Germany). These blots were in turn blocked at 37 °C for 1.5 h using an Odyssey Blocking Buffer (LI-COR Biosciences, USA). Then, the blots were incubated with anti-Nrf2 (1:1000), anti-Nucleus-Nrf2 (1:800), anti-Cytoplasm-Nrf2 (1:800), anti-HO-1 (1:1000), anti-SOD2 (1:1000), anti-NQO1 (1:1,000), anti-JNK (1:1000), anti-phospho-JNK (1:1000), anti-p38 (1:1000), anti-phospho-p38 (1:1000), anti-ERK (1:1000), and anti-phospho-ERK (1:1000) overnight at 4 °C for different experiments. For loading controls, blots were incubated with anti-β-actin (1:5000). After washing with 1×TBST for five times, blots were incubated with goat anti-rabbit IgG-HRP (1:5000) or goat anti-mouse IgG-HRP (1:5000) for 1 h at room temperature. After incubation, these blots were washed with with 1×TBST three times. A chemisope imaging system (CLiNX Science Instruments, Shanghai, China) was used for protein band detection. Samples were normalized to the Con group.

### 2.8. Statistical Analysis

Data are means ± standard errors (SEM). GraphPad Prism 5.0 (GraphPad, CA, USA) was used for all statistical testing. Data were compared via one-way ANOVAs with Tukey’s post-hoc test. P < 0.05 was the significance threshold.

## 3. Results

### 3.1. Quercetin Impacts on Intestinal Morphology of Broiler Chickens.

The intestinal morphology of broiler chickens in different groups is shown in Table 3. In the duodenum, jejunum, and ileum, LPS injection significantly decreased the villus height and increased the crypt depth (*p* < 0.05). In addition, LPS injection significantly decreased the ratio of villus height to crypt depth (VCR) (*p* < 0.05). However, a diet containing 200 mg/kg quercetin significantly alleviated these changes (*p* < 0.05). The 500 mg/kg Que+LPS group and the LPS group did not differ significantly (*p* > 0.05).

### 3.2. Effect of Quercetin on Oxidative Stress Induced by LPS

LPS challenge significantly increased ROS (Figure 1A), MDA (Figure 1B), and 8-OHdG (Figure 1C) levels (*p* < 0.05) and decreased GSH-Px (Figure 1E) and SOD (Figure 1F) activity; however, these were relieved by supplementation with 200 mg/kg of quercetin. However, there were no significant differences in CAT among the four groups, although the CAT value in the Que200+LPS group was higher than that in the LPS group. There was no significant differences between the Que500+LPS group and the LPS group (*p* > 0.05). Collectively, our data suggest that 200 mg/kg of quercetin could protect broiler chickens from the jejunal oxidative stress induced by LPS.

### 3.3. Effects of Quercetin on Jejunal Mitochondria of Broiler Chickens

Next, we observed the mitochondria by electron microscopy and quantified the relative mtDNA copy number. Compared to the Con group, there were alterations in the structure of the mitochondria of the LPS group (see arrows), including fusion of the internal structure and disappearance of mitochondria cristae, which were reversed in the LPS+Que group (Figure 2A). As shown in Figure 2B–D, quercetin also attenuated the mtDNA relative expression of ATP6, COX1, and ND1 reduced by LPS (*p* < 0.05).

### 3.4. Effects of Quercetin on Nrf2 Activation and Downstream Genes

To explore how Nrf2 signaling related to this process, we next assessed the impact of quercetin on the total level and nuclear localization of Nrf2 via Western blotting. As shown in Figure 3A, quercetin attenuated the inhibition of Nrf2 protein expression by LPS (*p* < 0.05). In addition, LPS treatment reduced the protein expression of N-Nrf2 (*p* < 0.05) and N-Nrf2/C-Nrf2 value (*p* < 0.01), and it upregulated the protein expression of C-Nrf2 (*p* < 0.05). However, compared with the LPS group, quercetin increased the protein expression of N-Nrf2 and the value of N-Nrf2/C-Nrf2, and it decreased the protein expression of C-Nrf2 (*p* < 0.05), indicating that quercetin decreased the nuclear translocation of Nrf2 from the cytoplasm to the nucleus. Subsequently, we examined the mRNA and protein levels of Nrf2 downstream genes and found that quercetin significantly attenuated the levels of manganese superoxide dismutase (SOD2), HO-1, and NQO1 reduced by LPS (Figure 3B–E) (*p* < 0.05).

### 3.5. Quercetin Impacts on ERK, JNK, and p38 Phosphorylation Reduced by LPS

Nrf2 activation and translocation could be activated by several signaling cascades. To further elucidate the upstream signaling pathways of Nrf2, we assessed ERK, JNK, and p38 phosphorylation. LPS significantly reduced the phosphorylation levels of ERK, JNK, and p38, while quercetin alleviated these reductions (*p* < 0.05; Figure 4). These results indicated that quercetin could increase the phosphorylation/activation of ERK, JNK, and p38.

## 4. Discussion

Maintaining gut health is important for the poultry industry. The ability of quercetin to protect against intestinal oxidative stress in domestic animals and poultry has rarely been reported. To the best of our knowledge, our results are the first demonstration that dietary quercetin could relieve LPS-induced intestinal morphological destruction and jejunal oxidative stress in broiler chickens, which is accompanied by destruction of jejunal mitochondrial structure and the decrease in mtDNA copy number. The MAPK/Nrf2 signaling pathway was involved in this process.

It has been widely demonstrated that LPS could damage the gut [6]. Accordingly, our results showed that LPS injection decreased the duodenal, jejunal, and ileal villus height while also increasing the crypt depth in these regions. We also found that the damage to the intestinal morphology was relieved by quercetin supplementation, indicating the protective effect of quercetin on intestinal injury. Previous studies support our results. For example, quercetin could alleviate intestinal injury in pigs caused by transport and was accompanied by the regulation of oxidative status and inflammation [20]. Similarly, quercetin could prevent intestinal damage during methotrexate-induced intestinal mucositis in rats [21]. These results indicate that quercetin could be a potential feed additive for broiler chickens to prevent intestinal injury induced by LPS injection.

Importantly, compared to the LPS group, quercetin reduced jejunal ROS production, downregulated MDA and 8-OHdG levels, and upregulated SOD and GSH-Px levels. SOD is an important scavenger for ROS. GSH-Px is also a primary readout for anti-peroxidative, and 8-OHdG is an indicator of DNA damage. The increase in lipid peroxidation levels and the decrease in antioxidant enzyme activity can lead to an imbalance between oxidation and antioxidants, triggering oxidative stress [22]. A previous study showed that quercetin could inhibit human intestinal oxidative stress by reducing ROS production [23]. Quercetin and selenium reduced MDA levels and inhibited oxidative stress caused by hydrogen peroxide and UV radiation in endometrial adenocarcinoma cells [24]. In the liver mitochondria, quercetin treatment decreased the ROS and MDA levels and increased the CAT and glutathione (GSH) levels [25]. Moreover, studies have reported that quercetin could be used as an effective agent to prevent mitochondrial dysfunction [26,27]. We also found that quercetin relieved mitochondrial morphological damage and increased the mitochondrial DNA copy number caused by LPS injection. Taken together, these results strongly suggested that quercetin inhibited the jejunal oxidative stress and mitochondrial damage induced upon LPS injection in broiler chickens.

Interestingly, comparing the results of two concentrations (200 or 500 mg/kg) of quercetin, we found that 200 mg/kg of quercetin had better antioxidant effects than 500 mg/kg. The reason may be that 500 mg/kg of quercetin is far beyond the effective range of quercetin to exert its role. Correspondingly, it has been demonstrated in other antioxidant studies that antioxidant effects are in a dose-dependent manner [28,29]. The present results will provide reference for the use of quercetin as a feed additive to protect broiler intestinal health.

How the Nrf2 signaling pathway functions in the inhibition of intestinal oxidative stress by quercetin remains undetermined. Previous studies have shown that quercetin could enhance the ARE (antioxidant response element) binding activity and Nrf2-mediated transcription activity to induce NQO1 expression in human HepG2 cells [30]. As found in macrophages, quercetin activated the Kelch-like ECH-associated protein-1(Keap1)–Nrf2 signaling pathway to induce HO-1 expression for protection against oxidative stress and inflammatory responses [10]. Quercetin could effectively inhibit manganese-induced oxidative stress and inflammatory response in neuraltumor epithelialcells (SK-N-MC) in Sprague–Dawley rats though activating HO-1/Nrf2 and inhibiting the NF-κB pathway [31]. Additionally, quercetin could improve mitochondrial function by promoting the translocation of Nrf2 from the cytoplasm to the nucleus and activating the Nrf2 signaling pathway [32,33]. In our study, quercetin promoted Nrf2 levels and translocation, and it increased Nrf2 downstream genes levels (HO-1, NQO1, and SOD), suggesting that quercetin inhibits LPS-induced intestinal oxidative stress through the Nrf2 signaling pathway. In addition, MAPK plays an important role in various extracellular signals through the cascade of continuous phosphorylation and is key mediator of the activation and translocation of Nrf2 [34,35,36]. Our results showed that quercetin relieves the inhibition of ERK, JNK, and p38 phosphorylation induced by LPS injection, indicating that the MAPK/Nrf2 pathway may be involved in mediating the protective effect of quercetin. In human hepatocytes, quercetin inhibited ethanol-derived oxidative stress via the MAPK/Nrf2 pathway. This pathway was also involved in quercetin inhibition of the inflammatory response in BV-2 microglia [37]. Quercetin could also activate the p38 or ERK pathways and induce HO-1 expression to prevent H_2_O_2_-induced apoptosis [38,39]. These results support our finding that quercetin exerted antioxidant effects via the Nrf2 signaling pathway.

## 5. Conclusions

In conclusion, our results demonstrate that quercetin could prevent LPS injection-induced jejunum injury in broiler chickens by relieving oxidative stress via the activation of the Nrf2 signaling pathway (Figure 5). Therefore, our findings provide strong evidence for quercetin as an effective feed additive to protect intestinal health.

## Figures and Tables

**Figure 1 molecules-25-01053-f001:**
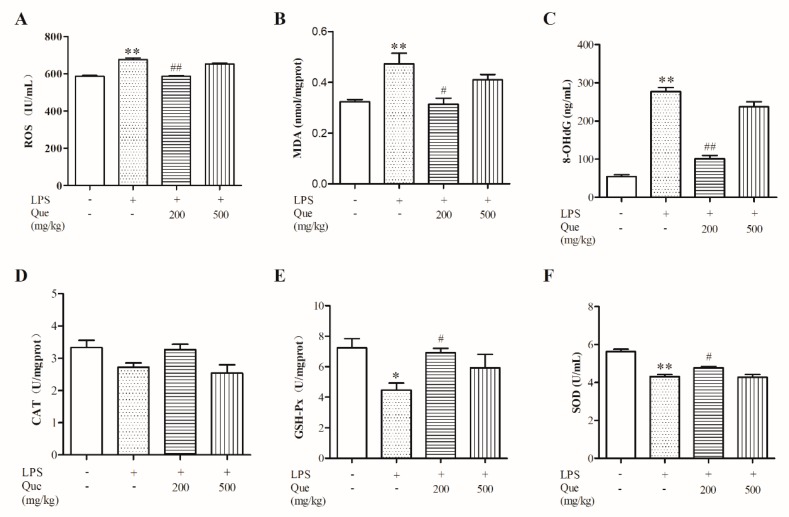
The effect of quercetin on oxidative stress induced by lipopolysaccharide (LPS). The levels of reactive oxygen species (ROS) (**A**), malondialdehyde (MDA) (**B**), 8-hydroxy-2′-deoxyguanosine (8-OHdG) (**C**), catalase (CAT) (**D**), glutathione peroxidase (GSH-Px) (**E**), and superoxide dismutase (SOD) (**F**) were measured by ELISA or chemical colorimetry (n = 6). Values are shown as the mean ± SEM. ** *p* < 0.01 or * *p* < 0.05 compared with the Con group. ## *p* < 0.01 or # *p* < 0.05 compared with the LPS group.

**Figure 2 molecules-25-01053-f002:**
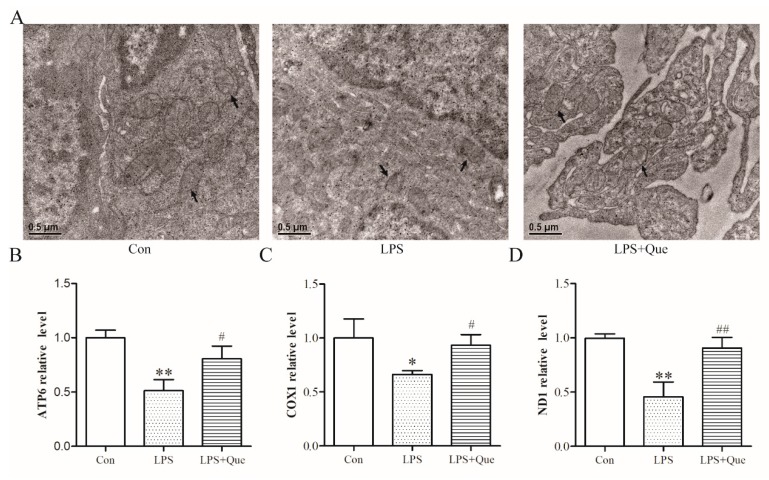
The effects of quercetin on jejunal mitochondria of broiler chickens. (**A**) Mitochondria from different groups were observed by electron microscopy. The mtDNA relative expression levels of ATP synthase F0 subunit 6 (ATP-6) (**B**), cytochrome c oxidase subunit 1 (COX1) (**C**), and NADH dehydrogenase subunit 1 (ND1) (**D**). Values are shown as the mean ± SEM. ** *p* < 0.01 or * *p* < 0.05 compared with the Con group. ## *p* < 0.01 or # *p* < 0.05 compared with the LPS group.

**Figure 3 molecules-25-01053-f003:**
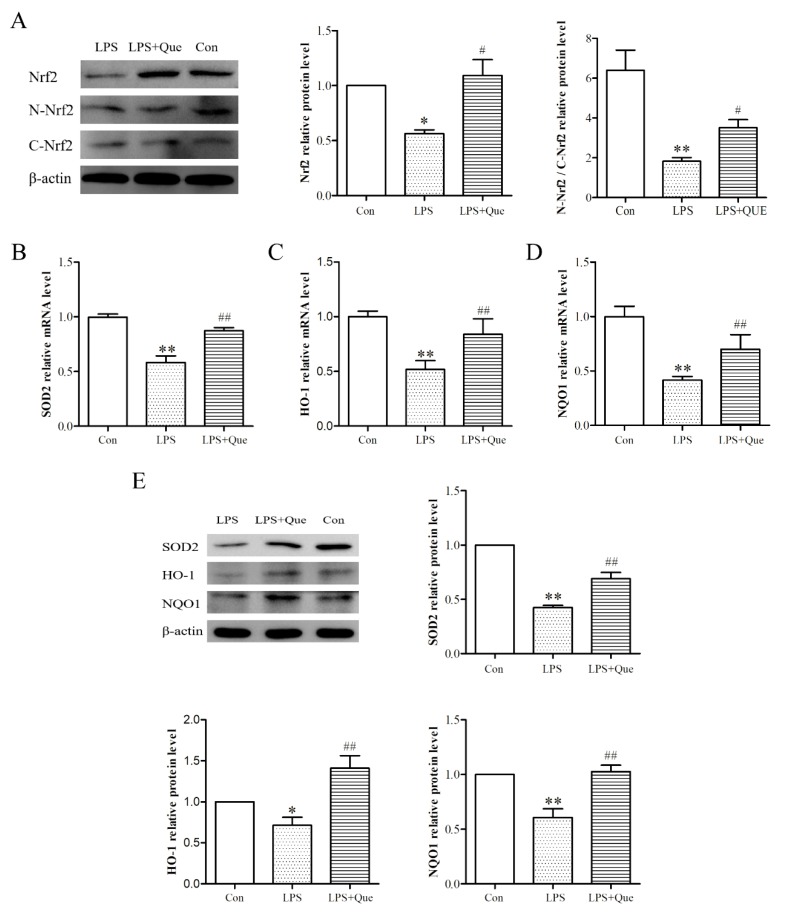
The effects of quercetin on nuclear factor erythroid 2-related factor 2 (Nrf2) activation and downstream genes. (**A**) The relative protein levels of Nrf2, N-Nrf2, and C-Nrf2. The relative mRNA levels of manganese superoxide dismutase (SOD2) (**B**), heme oxygenase-1 (HO-1) (**C**), and NAD(P)H dehydrogenase quinone 1 (NQO1) (**D**). (**E**) The relative protein levels of SOD2, HO-1, and NQO1. Values are shown as mean ± SEM. ** *p* < 0.01 or * *p* < 0.05 compared with the Con group. ## *p* < 0.01 or # *p* < 0.05 compared with the LPS group.

**Figure 4 molecules-25-01053-f004:**
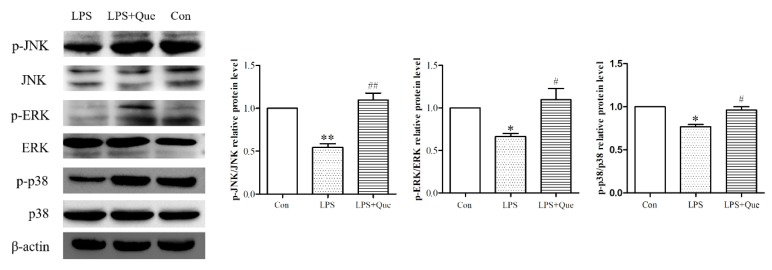
The effects of quercetin on total and phosphorylated levels of ERK and JNK. Total and phosphorylated levels of ERK, JNK, and p38 in different groups measured by Western blot. Values are shown as mean ± SEM. ** *p* < 0.01 or * *p* < 0.05 compared with the Con group. ## *p* < 0.01 or # *p* < 0.05 compared with the LPS group.

**Figure 5 molecules-25-01053-f005:**
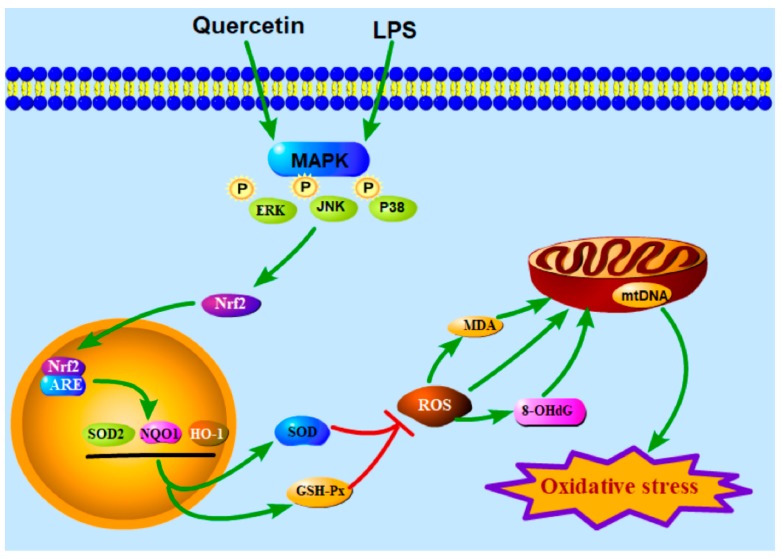
A scheme depicting the effects of quercetin activating the Nrf2 pathways to inhibit LPS-induced intestinal oxidative stress.

**Table 1 molecules-25-01053-t001:** Ingredients and nutrient composition of the basal diet.

Ingredients (g/kg)	Values	Calculation of nutrients (g/kg)^y^	Values
Corn	586.267	Crude protein	21
Soya bean meal (46%)	320.722	Metabolism energy (MJ/kg)	3050
Soybean oil	35.259	Moisture	12.3807
Corn gluten meal	20	Dry matter	87.6193
Limestone	11.712	Crude fat	6.1026
DL-methionine (98%)	2.301	Crude fiber	3.1719
Lysine (98%)	3.829	Crude ash	5.5485
L-threonine	0.525	Calcium	0.9
Choline chloride (60%)	1	Total phosphorus	0.6361
calcium hydrogen phosphate	10.315	Available phosphorus	0.41
Salt	3.32	Sodium	0.1492
Vitamin-Trace mineral premix ^z^	3.7	Chlorine	0.2266
Phytase (10000 units)	0.1	Lysine	1.35
L-tryptophan (20%)	0.95	Methionine	0.55
		Sulfur-containing amino acid	0.878
		Threonine	0.83
Total	1000	Tryptophan	0.25

^z^ Vitamin for broiler chickens provided per kilogram of diet: VA, 10,000 IU; VB_1_, 2 mg; VB_2_, 7 mg; VB_6_, 4 mg; VB_12_, 0.02 mg; VD, 4000 IU; VE, 25 mg; VK, 2 mg; biotin, 0.1 mg; folic acid, 1.2 mg; niacinamide, 40 mg; calcium pantothenate, 10 mg. Fe (from ferrous sulfate), 100 mg; Zn (from zinc oxide), 65 mg; Cu (from copper sulfate), 10 mg; Mn (from manganese sulfate), 100 mg; Se (from sodium selenite), 0.3 mg; I (from calcium iodate), 0.7 mg; ^y^ Crude protein was a measured value, while the others were calculated values.

**Table 2 molecules-25-01053-t002:** Primer sequences used in the present study.

Genes	Primer Sequence (5′– 3′)	Size (bp)	Tm (°C)
HO-1	F: AAGAGCCAGGAGAACGGTCA	121	57
	R: AAGAGCCAGGAGAACGGTCA		
SOD2	F: CTTGGTCGCAAGGCAGAAG	120	57
	R: ACGTAGGTGGCGTGGTGTT		
NQO1	F: TCGCCGAGCAGAAGAAGATTGAAG	192	57
	R: CGGTGGTGAGTGACAGCATGG		
β-actin	F: GTGCTATGTTGCTCTAGACTTCG	174	57
	R: ATGCCACAGGATTCCATACC		
ND1	F: GCGCCCCATTTGACCTAACA	85	58
	R: AATATGGCGAATGGTCCGGC		
COX1	F: TCCTTACCCGTCCTAGCAGC	134	58
	R: TCGGGGTGACCGAAGAATCA		
ATP6	F: GATCAACAACCGCCTCTCCA	111	58
	R: GAGGTGAGTAGGAGGGCTCA		
AGRT	F: TGGCCATAGTGCATCCAGTG	199	58
	R: ACGATGAATGATGACGGGCA		

**Table 3 molecules-25-01053-t003:** Intestinal morphology of broiler chickens in the four groups.^z^

Item	Con^y^	LPS^x^	200 mg/kg Que + LPS^w^	500 mg/kg Que + LPS^v^
Duodenum				
Villi height (μm)	718.1±29.1^a^	582.8±23.3^b^	749.6±26.9^a^	566.4±8.7^b^
Crypt depth (μm)	103.1±10.6^b^	155.5±9.4^a^	104.6±11.2^b^	139.7±13.5^a^
Jejunum				
Villi height (μm)	414.4±8.5^b^	362.9±6.5^c^	463.9±4.5^a^	338.9±13.5^c^
Crypt depth (μm)	60.4±7.8^b^	85.3±5.6^a^	70.5±9.1^b^	86.7±10.5^a^
Ileum				
Villi height (μm)	413.7±16.7^a^	325.0±10.6^b^	356.0±13.6^a^	304.8±11.9^b^
Crypt depth (μm)	50.2±6.1^c^	86.4±5.2^a^	54.2±7.5^c^	68.1±9.5^b^

^z^ Values are given as means based on six birds (one bird per replicate); ^y^ = basal diet + saline challenge; ^x^ = basal diet + LPS challenge; ^w^ = basal diet with 200 mg/kg Que + LPS challenge; ^v^ = basal diet with 500 mg/kg Que + LPS challenge; ^a,b,c^ Means in a row bearing different superscripts are significantly different (*p* < 0.05).

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
