# Peer review of "Quercetin Protects against Lipopolysaccharide-Induced Intestinal Oxidative Stress in Broiler Chickens through Activation of Nrf2 Pathway"

_molecules, 2020, doi:10.3390/molecules25051053_

Round 1

Reviewer 1 Report

The Authors investigated the potential role of orally administered quercetin in counteracting the intestinal oxidative stress induced by LPS in broilers. Amelioration of intestinal morphology and antioxidant parameters (i.e. GPx and SOD activity), as well as reduction of oxidative indicators (e.g. ROS and MDA levels), in the presence of quercetin are reported. Furthermore, the Authors explored the possible involvement of the MAPK/Nrf2 pathway in the quercetin-driven protective effect.
From a general point of view, the subject of the study is worth of investigation and the research is well-designed and performed with appropriate techniques. Moreover, the results are robust and interesting, providing new insights into the possible use of quercetin as a feed additive to protect broiler intestinal health. However, there are some remarks that should be considered before publishing.
INTRODUCTION.
1. Lines 48-49: I suggest the Authors revise “oxidative stress state” with “antioxidant status”. In my opinion it is more correct.
2. Lines 51-52: The sentence is not clear. Please, rephrase.
M&M.
1. The Authors describe the experimental design including 240 animals. However, only 6 animals per treatment were employed. Thus, in my opinion the Authors should underline such condition in section 1.1 for the sake of clarity. Similarly, I suggest the Authors insert the number of animals (n=) in each figure caption.
2. Lines 68-69: the employed dosage is not clear. Is it expressed as a concentration in water or as a quantity per b.w.? Please, revise.
3. Lines 96-97: The sentence is not clear. Please, rephrase.
4. Section 2.4: Could the Authors provide more information about the assays they performed? For example, were they colorimetric or fluorimetric tests?
5. In sections 2.5 and 2.7 it is not reported on which intestinal part the analyses were performed. Please, insert such information.
6. In the results the Authors report the effects on mitochondria as investigated by electron microscopy. However, such analysis has not been described in the M&M. Please, include it in section 2.2 or in a new one.
7. Table 1: What does the “1-21 days” refer to? If it is not necessary, it could be deleted.
RESULTS.
1. Line 159: the results related to CAT activity are not statistically significant. Thus, in my opinion the Authors should revise accordingly both the results and the discussion.
2. Figure 1 legend: “measured by a microplate reader” is not useful in the absence of information about the assay principles.
3. Starting from section 3.3 the Authors do not report the investigated quercetin concentration (200 or 500?). They only refer to LPS+Que group. If they investigated only one group, they should indicate the concentration and substantiate their choice.
4. Lines 184-187: In my opinion the sentence is quite confusing. The Nrf2 translocation from the cytoplasm to the nucleus activates the antioxidant defence inducing the transcription of target genes (e.g. SOD, NQO1...). Thus, the higher level in the nucleus is a positive effect, as depicted in Figure 3A. The Authors should rewrite such paragraph according to their results. I also suggest the substitution of the verb “attenuated” since it does not clearly describe the protective effect of quercetin.
DISCUSSION
Please, for the sake of clarity, rephrase lines 241-243 and lines 251-253.

Reviewer 2 Report

In their manuscript Sun et al., analyzed the potential application of quercitin to protect broiler chicken against LPS-induced intestinal oxidative stress. They found that as previously shown quercetin effect is related to its ability to reduce LPS-induced oxidative stress via the MAPK/Nrf2 signaling pathway. In general the manuscript is clear and well presented, my only concern regards the novelty of the data presented since the ability of quercitin to ameliorates gut inflammation it has been shown in other animal model.
